# A Comprehensive Review of the Genetics of Dyslipidemias and Risk of Atherosclerotic Cardiovascular Disease

**DOI:** 10.3390/nu17040659

**Published:** 2025-02-12

**Authors:** Megan Kalwick, Mendel Roth

**Affiliations:** GBinsight, GB Healthwatch, San Diego, CA 92122, USA; megan.kalwick@gblifesciences.com

**Keywords:** dyslipidemia, hypercholesterolemia, hypertriglyceridemia, hypoalphalipoproteinemia, hyperalphalipoproteinemia, Lp(a), genetics, atherosclerotic cardiovascular disease

## Abstract

Dyslipidemias are often diagnosed based on an individual’s lipid panel that may or may not include Lp(a) or apoB. But these values alone omit key information that can underestimate risk and misdiagnose disease, which leads to imprecise medical therapies that reduce efficacy with unnecessary adverse events. For example, knowing whether an individual’s dyslipidemia is monogenic can granularly inform risk and create opportunities for precision therapeutics. This review explores the canonical and non-canonical causes of dyslipidemias and how they impact atherosclerotic cardiovascular disease (ASCVD) risk. This review emphasizes the multitude of genetic causes that cause primary hypercholesterolemia, hypertriglyceridemia, and low or elevated high-density lipoprotein (HDL)-cholesterol levels. Within each of these sections, this review will explore the evidence linking these genetic conditions with ASCVD risk. Where applicable, this review will summarize approved therapies for a particular genetic condition.

## 1. Introduction

Dyslipidemias are a major, causal risk factor for atherosclerotic cardiovascular disease (ASCVD) and other chronic diseases. Primary causes of dyslipidemias are genetic, while secondary causes arise due to other factors such as type 2 diabetes, thyroid disorders, or poor diet. In this review, we will focus on primary causes of dyslipidemias and highlight its often-cited and less-publicized causes. And, when appropriate, we will highlight precision therapies that are more suitable for one genetic cause over another. This review will discuss in some detail the evidence, when applicable, linking a particular gene, genetic variant, or genetic condition to the risk of ASCVD.

ASCVD is a complex disease with a myriad of inputs in diverse pathways. Often, these different pathways additively interact to contribute to a greater risk of ASCVD. The higher number of these affected pathways with longer exposures and greater magnitudes of the insults leads to an earlier age of onset. For example, people with homozygous FH (familial hypercholesterolemia) begin to develop plaque within their first decade of life, and, if untreated, can have a cardiovascular disease event within their second decade of life. Thus, genetic conditions portend a greater risk because the exposures occur since gestation.

Comprehensive genetic analysis using next-generation sequencing, such as the GBinsight Dyslipidemia and ASCVD panel, is designed to comprehensively analyze the multitude of pathways contributing to ASCVD risk. Figure 1 shows these pathways, and a selection of genes analyzed. This review expounds on these pathways and discusses the evidence supporting their relationship to ASCVD risk. This review is divided into three major sections focused on the major dyslipidemia classes: hypercholesterolemia, hypertriglyceridemia, and HDL-cholesterol abnormalities (low and high HDL-C).

## 2. Hypercholesterolemia

Background: There are familial and acquired causes of hypercholesterolemia. This review will focus on the multitude of genetic causes of hypercholesterolemia. Since Michael Brown and Joseph Goldstein identified that genetic mutations within the LDL receptor gene (LDLR) cause FH, the cholesterol clearance pathway is recognized as the canonical genetic cause of primary hypercholesterolemia. Many papers have been written about FH, and so this review will summarize this pathway but refer readers to other reviews that offer more details [1].

Monogenic FH: Monogenic FH is an autosomal co-dominant condition and is the most common Mendelian disease with a prevalence of 1:250 globally but can be as frequent as 1:100 in some founder populations [2]. Pathogenic variants within the LDLR that encode the LDL receptor (LDLr) account for approximately 90–95% of monogenic FH. There are nearly 2500 pathogenic/likely pathogenic variants reported in the LDLR, according to Clinvar. Rare missense variants within the APOB gene that prevent binding of apoB-100 to the LDLr account for 5% of monogenic FH; Arg3527Gln (also known as Arg3500Gln or R3500Q) is the prototypical FH-causing pathogenic variant. This subtype is also known as familial defective apolipoprotein B-100 (FDB). Pathogenic gain-of-function (GOF) variants within the PCSK9 gene account for 1% of monogenic FH.

Between 40 and 60% of people with clinical FH have a detectable pathogenic/likely pathogenic variant in one of the canonical FH genes [3,4]. And this percentage increases linearly to more than 90% as serum low-density lipoprotein cholesterol (LDL-C) levels exceed 300 mg/dL [5].

Several independent studies demonstrate the clinical utility of knowing whether a patient has a monogenic cause of FH or another cause of clinical FH. People with monogenic FH are three-fold more likely to have ASCVD compared to non-carriers. This enhanced risk is independent of LDL-C strata [6,7,8,9]. This is likely due to the persistent severe hypercholesterolemia occurring since fetal development. Studies further show that people with null LDLR alleles, namely, those that cause splicing defects or a truncated protein are at an even greater ASCVD risk [10,11]. In effect, there is a dose–response relationship conferring increased ASCVD risk in people with a longer duration of exposure and greater magnitude of hypercholesterolemia.

Additionally, people with a heterozygous FH-causing pathogenic variant have a 50% chance of transmitting this pathogenic variant to their offspring. Public health agencies in the United States and Europe assign Tier 1 recommendations for genetic cascade screening for HeFH (US CDC and UK NICE). This is because the earlier detection of HeFH results in earlier treatment, which saves lives [12].

LDLRAP1: A rare cause of autosomal recessive hypercholesterolemia (ARH) is caused by pathogenic, loss-of-function variants in the LDLRAP1 gene. This gene, located on human chromosome 1p36.11, encodes for the LDL receptor adaptor protein 1 that helps facilitate the endocytosis of LDL-LDL receptor complexes. With genetic functional absence of the LDLRAP1 protein, the effective clearing of LDL particles is greatly impaired, resulting in LDL-C levels within the HoFH range (>400 mg/dL). Because ARH is a subtype of HoFH, treatment for people with ARH includes the ANGPTL3 inhibitor, evinacumab, on top of traditional lipid-lowering therapies.

### 2.1. Autosomal Dominant Hypercholesterolemia (ADH): APOE

The APOE gene, residing on human chromosome 19q13.32, encodes a small apolipoprotein, apoE, that has many functions in lipid biology. ApoE is produced and secreted into the peripheral circulation by hepatocytes where it incorporates into most classes of lipoproteins, but preferentially to very low-density lipoprotein (VLDL) and chylomicron remnants. In the periphery, apoE is primarily involved in the clearance of triglyceride-rich lipoprotein remnants. As such, genetic loss-of-function variants, including people that are homozygous for the APOE2 allele, are at risk for developing familial dysbetalipoproteinemia (FDBL, also known as type III hyperlipoproteinemia) that is clinically characterized by very high lipoprotein remnant concentrations and hypertriglyceridemia.

An in-frame deletion of Leucine167 (c.500_502del(p.Leu167del)) within the LDL receptor-binding region of apoE is known to be pathogenic for ADH [13,14,15]. This variant disrupts the structure of the LDL receptor-binding domain within apoE. This variant was the most common cause of monogenic hypercholesterolemia in people without FH-causing variants in the LDLR, APOB, or PCSK9 genes [16].

Another variant in the APOE gene that is colloquially known as E4-Freiburg has been shown to be associated with robust increases in cholesterol levels, according to data from the UKBiobank [17,18]. APOE4-Freiburg refers to a missense variant, c.137T>C(p.Leu46Pro) (also known as Leu28Pro or L28P), occurring on the same chromosomal copy (in cis) with the APOE4 allele (p.Cys130Arg). Several groups have reported this variant as likely pathogenic for ADH [19,20]. Molecular studies show that Leucine46 resides in the first α-helix of the N-terminal domain, and the substitution for proline (Leu46Pro) destabilizes this domain, leading to enhanced proteolysis and reduced lipid binding [19].

### 2.2. Phenocopying FH

ABCG5/8: The ABCG5 and ABCG8 genes, located back-to-back on human chromosome 2p21, encode two proteins that heterodimerize to form sterolin, a molecular pump expressed in the small intestine and liver. On enterocytes, sterolin pumps absorbed sterols (primarily plant sterols but also cholesterol) back into the gut lumen. On hepatocytes, sterolin pumps sterols out into gallbladder canaliculi for excretion [21]. Bi-allelic genetic loss of ABCG5/8 causes sitosterolemia, which is characterized by high plasma phytosterols, hypercholesterolemia, xanthomata, premature atherosclerosis, and hematological sequelae. First-line therapy for sitosterolemia is ezetimibe that blocks gut absorption and hepatic reuptake of sterols from the gallbladder by inhibiting the NPC1L1 sterol transporter.

Heterozygous carriers of pathogenic variants in these genes, but especially ABCG5, have moderate hypercholesterolemia and an increased risk of ASCVD [21,22]. Unpublished data suggest that heterozygosity for pathogenic ABCG5/8 variants may explain why some people who adhere to very low-carbohydrate/ketogenic diets that are abundant in dietary cholesterol can see dramatic spikes in serum cholesterol levels.

In some cohorts, approximately 3% of people with clinically diagnosed FH are carriers of ABCG5/8 pathogenic variants [23,24,25], and including these genes in FH genetic testing panels increases yield and has precision medicine implications. A retrospective analysis of people with clinical FH shows that ezetimibe (on top of statins) shows an improved lipid-lowering profile in people with ABCG5/8 pathogenic variants compared to non-carriers [26].

LIPA: The LIPA gene resides on human chromosome 10q23.31 and encodes the enzyme lysosomal acid lipase (LAL). As the name indicates, this enzyme is active in lysosomes and is involved in the breakdown of lipids from the core of endocytosed lipoproteins. This reaction releases free cholesterol and fatty acids as part of the catabolism of these lipoproteins.

Bi-allelic genetic loss of the LIPA gene causes lysosomal acid lipase deficiency (LALD). There are two subtypes of LALD: the more severe form, called Wolman disease is due to genetic variants that result in the complete absence of lysosomal acid lipase (LAL) activity; the other subtype is known as cholesteryl ester storage disease (CESD) and results from genetic variants causing a partial loss of LAL activity. Wolman disease manifests in infancy causing failure to thrive, hepatomegaly, splenomegaly, adrenal insufficiency, and death in early childhood, when untreated. Partial LALD is characterized by splenomegaly, hepatomegaly with fibrosis, and a high risk of ASCVD. A lipid panel in someone with LALD shows hypercholesterolemia, hypertriglyceridemia, and low HDL-C. Enzyme replacement therapy with sebelipase alfa is currently the only approved treatment for people with LALD.

Heterozygous carriers are not known to have severe dyslipidemia but may have moderate hypercholesterolemia and moderate hypertriglyceridemia. Other genetic and non-genetic factors may interact with LIPA heterozygous loss-of-function genetic variants to cause clinically relevant dyslipidemia.

ALB: A very rare cause of severe hypercholesterolemia is analbuminemia, an autosomal recessive disease due to genetic loss of the ALB gene that encodes for albumin. Analbuminemia is a very rare disease with an estimated prevalence of 1:1,000,000. Clinically, analbuminemia is characterized by very low plasma albumin, hypotension, and severe hypercholesterolemia but normal triglycerides and HDL-cholesterol. The infant mortality rate is high [27].

High Lp(a): Lp(a) is a derivative of LDL with one molecule of apolipoprotein(a) (apo(a)) covalently bound to apoB-100 on LDL. The LPA gene resides on human chromosome 6q25 and encodes for the apo(a) protein. The LPA gene is remarkable, as it is present in only a few other animals besides humans, including Old World nonhuman primates and the European hedgehog [28]. The LPA gene itself arose due to a duplication of the plasminogen (PLG) gene, sharing strong homology of 70% of genetic sequences and nearly 100% of parts of the encoded apo(a) protein. This has clinical implications as Lp(a), due to its apo(a) moiety, has pro-thrombotic properties, which helps make Lp(a) 6x more atherogenic than LDL at the molar level [29].

The reason for the increased atherogenicity of Lp(a) is several-fold: (i) The LDL moiety is atherogenic. (ii) Lp(a) preferentially carries oxidized phospholipids (OxPLs) within the Kringle-IV, type 10 (KIV-10) domain, and OxPLs are pro-inflammatory that promotes endothelial dysfunction [30]. We refer the reader to another review on the role of OxPLs in Lp(a) atherogenicity [30]. (iii) Due to its homology with plasminogen, Lp(a) may promote thrombosis by inhibiting the activation of plasminogen and therefore limits fibrinolysis [31].

Some studies have demonstrated that one common genetic variant that serves as a genetic proxy for short isoforms in Caucasian populations, rs3798220, may uniquely promote thrombosis [32,33]. Interestingly, post hoc analysis from two separate randomized controlled trials show that people with this genetic variant are more amenable to anti-platelet aspirin therapy [34,35]. In the recent post hoc analysis of the ASPREE trial by Lacaze et al. (2022), the authors suggest that people with a high polygenic risk score (PRS) and not just carriers of the rs3798220-C allele may benefit from aspirin. However, these authors employ a PRS that already incorporates the rs3798220 variant and the hazard ratio for the high Lp(a) PRS group taking aspirin is attenuated compared to the rs3798220-C carriers. This suggests that the cardioprotection due to aspirin is likely due to the rs3798220 variant rather than due to high Lp(a) more broadly.

High Lp(a), defined as greater than 50 mg/dL or 125 nmol/L, is associated with an increased risk of ASCVD, calcific aortic valve stenosis, and, to a lesser extent, other cardiovascular diseases [36]. There is a linear relationship between Lp(a) levels and ASCVD risk [37]. High Lp(a) can be considered monogenic as the LPA gene is the primary driver of Lp(a) levels, accounting for more than 90% of its population variance [38,39]. A two-exon variable number tandem repeat (VNTR) that corresponds to the Kringle-IV, type 2 (KIV-2) domain of apo(a) is the major genetic cause of Lp(a) variability. There is an inverse relationship between the number of these unit repeats and serum Lp(a) levels. Across the general population, people have between 2 and 50 of these repeats. Apo(a) with a small number of KIV-2 repeats encodes for short isoforms, while apo(a) with a larger number of repeats encodes for longer isoforms. The molecular explanation for this inverse relationship is that very large apo(a) is not produced as efficiently in hepatocytes, and the very large nascent peptide is less stable [40], resulting in low serum Lp(a) levels.

Since Lp(a) carries cholesterol in the LDL portion, people with high Lp(a) may present with hypercholesterolemia. And because high Lp(a) is familial and increases ASCVD risk, people with very high Lp(a) can present with clinical FH. Several studies show that people with clinical FH have higher Lp(a) levels compared to non-FH [41,42,43]. We refer the reader to the following article for more information about this [44].

While a discussion on treatments for high Lp(a) is beyond the scope of this review, it is important to point out that, even though targeted therapies for treating high Lp(a) are currently lacking, it is incomplete to conclude that there is nothing that currently can be performed for patients with high Lp(a) at a high ASCVD risk. From the prospective EPIC-Norfolk study, subjects with high Lp(a) who abided by a healthy lifestyle were much less likely to develop cardiovascular disease compared to those with high Lp(a) with an unhealthy lifestyle (hazard ratio, HR = 0.33 (95% CI = 0.17–0.63)) [45].

### 2.3. Polygenic FH

Polygenic FH refers to people without a pathogenic FH-causing variant but instead have a large burden of genetic variants that individually confer a small-to-moderate increase in serum cholesterol levels and who have few cholesterol-lowering variants. Studies show that, of people with clinical FH but without a monogenic variant identified, there is an overrepresentation of people with a high polygenic FH score [3,46]. The APOE4 allele is the most highly weighted in polygenic FH scores. As noted, high Lp(a) may contribute to polygenic FH. Lp(a), being a derivative of LDL, carries cholesterol and so people with very high Lp(a) may have higher blood cholesterol levels, but this is largely contained within their Lp(a), designated as Lp(a)-C. Other single nucleotide polymorphisms (SNPs) within hypercholesterolemia-associated genes, such as LDLR, APOB, ABCG8, etc., also contribute to polygenic FH.

### 2.4. FH Candidate Genes

STAP1: Researchers, looking for other causes of FH, reported in the mid-2000s another gene suspected to cause autosomal dominant hypercholesterolemia and was given the moniker ADH4 [47]. The gene, STAP1, encodes for an adaptor protein that mediates signals from outside the cell to inside. Subsequent studies in model rodents and genomic studies in humans led to the conclusion that STAP1 does not cause ADH [48,49,50].

SORT1: Multi-ethnic genome-wide association studies consistently identified signals in chromosomal region 1p13.3 associated with hypercholesterolemia and ASCVD risk [51,52,53,54,55]. There are several genes within this chromosomal region: CELSR2-PSRC1-SORT1-MYBPHL. Of these genes, SORT1 is the most biologically plausible. The SORT1 gene encodes sortilin-1 that is involved in the endocytic vesicle trafficking pathway. Subsequent studies showed that sortilin-1 is expressed in hepatocytes and is involved in LDL clearance [56].

Molecular genetics studies further revealed that the alleles associated with reduced cholesterol levels and lower ASCVD risk specifically increase sortilin-1 gene expression in hepatocytes and that is associated with suppressed apoB-100 and VLDL secretion in the liver as well as enhanced LDL uptake [57,58,59].

Human genetics studies are conflicting and remain inconclusive whether rare SORT1 genetic variants cause ADH. In one study, rare SORT1 coding variants identified in people with hypercholesterolemia did not appreciably affect LDL dynamics [60]. In another study of Amish individuals in the US, missense genetic variants were functionally associated with cholesterol levels [61]. Gene burden data from the UKBiobank that collates loss-of-function (LOF) SORT1 variants do not support an association with LDL-C levels [62].

MYLIP: The MYLIP gene, located on human chromosome 6p22.3, encodes an E3-ubiquitin ligase, inducible degrader of the low-density lipoprotein receptor (IDOL), that is involved in the proteasome-mediated degradation of the LDL receptor. This candidate gene has biological plausibility to be an FH-candidate gene as the encoded protein (IDOL) is involved in the degradation of the LDL receptor. A common SNP within the MYLIP gene, p.Asn342Ser (also known as N342S; rs9370867), is associated with serum cholesterol levels [63,64]. Rare genetic variants have also been shown in people with hypercholesterolemia without a known genetic cause [65,66]. For example, the missense variant, p.Gly51Ser, was found in families with hypercholesterolemia, and functional studies show this variant enhances the degradation of the LDL receptor [66]. It is therefore possible that genetic gain-of-function variants lead to hypercholesterolemia by promoting the proteasomal degradation of the LDL receptor. Further studies are needed to establish the MYLIP gene as a cause of ADH.

## 3. Hypertriglyceridemia

Background: Moderate hypertriglyceridemia, but more precisely, triglyceride-rich lipoprotein (TRL) is associated with increased ASCVD risk, according to epidemiological and genetic studies [10,67,68,69]. Unlike with serum cholesterol levels, there is no available compelling pharmacological evidence that lowering serum triglyceride levels reduces ASCVD risk, raising some doubt about whether hypertriglyceridemia is proximally causal for ASCVD.

Primary hypertriglyceridemia is largely polygenic with both genetic and non-genetic factors involved and is associated with low HDL-C, insulin resistance, and metabolic syndrome. Severe hypertriglyceridemia (sHTG) can cause acute pancreatitis (AP) and insulin resistance. There is significant morbidity and mortality associated with AP, which underscores the need to understand the proximate cause and treat accordingly. There is some preliminary evidence that responsiveness to a newer class of triglyceride-lowering therapies depends on whether someone has familial chylomicronemia syndrome (FCS) or multifactorial chylomicronemia syndrome (MCS). Briefly, people with FCS do not respond as well to anti-ANGPTL3 monoclonal antibody therapy compared to MCS [70].

The canonical cause of sHTG is due to genetic defects in the catabolism of TRLs, notably chylomicrons. FCS is an umbrella term for several conditions that result in defects in lipoprotein lipase (LPL) from hydrolyzing triglycerides and phospholipids from TRLs for uptake by peripheral tissues. FCS is associated with significant clinical risk for AP but is not believed to increase the ASCVD risk as these lipoprotein particles are too big to traverse the endothelium [71].

Familial chylomicronemia syndrome: FCS is a very rare condition with an estimated prevalence of 1:250,000. FCS is an autosomal recessive disease characterized by severe hypertriglyceridemia, recurrent episodes of acute pancreatitis, eruptive xanthomas, lipemia retinalis, insulin resistance, and metabolic dysfunction-associated fatty liver disease (MAFLD). The major genetic cause of FCS is bi-allelic pathogenic variants within the LPL gene, accounting for 80% of all FCS cases. Other genes causing syndromes that are clinically similar to FCS include APOA5, APOC2, LMF1, and GPIHBP1.

LPL: The LPL gene, located on human chromosome 8p21.3, encodes for lipoprotein lipase that is produced primarily by white adipocytes and skeletal myocytes and anchored to endothelial membranes where it docks with TRLs and hydrolyzes triglycerides and, in the process, produces remnant lipoproteins.

Heterozygous carriage of loss-of-function LPL genetic variants increases the ASCVD risk [72]. Consistently, genetic variants within LPL that reduce triglyceride levels are associated with reduced ASCVD comparable to LDL-C-lowering LDLR variants [73]. This further lends credence to the causality of moderate hypertriglyceridemia and increased TRLs for ASCVD.

There is some controversy whether heterozygosity for LPL loss-of-function variants is sufficient to cause moderate-to-severe hypertriglyceridemia. Longitudinal studies of heterozygous LPL pathogenic variant carriers show large variability within and between subjects [74]. Nearly two-thirds of TG measurements in this cohort of 15 individuals were in the moderately elevated range, while 18% of these measurements were reflective of sHTG. It is likely that the type of LPL variant influences the severity of hypertriglyceridemia: a null allele without any residual activity is likely to have a more severe phenotype compared to a missense LPL pathogenic variant with 25% residual activity. Additionally, non-genetic factors, such as diet and alcohol consumption, likely influence the risk of sHTG.

APOA5: The APOA5 gene, located on human chromosome 11q23.3, encodes the apoA-V protein that serves as an activator of lipoprotein lipase. Bi-allelic genetic LOF APOA5 variants cause FCS. Heterozygous carriage of LOF APOA5 variants can cause moderate HTG and is associated with an increased ASCVD risk [10]. Like with LPL, there is variability in the severity of HTG in heterozygous carriers. This likely has to do with the type of variant and other non-genetic factors.

APOC2: The APOC2 gene, located on human chromosome 19q13.32, encodes the apoC-II protein that is a requisite cofactor for the activation of lipoprotein lipase. FCS due to bi-allelic genetic loss of APOC2 is an extremely rare subtype. To date, there are no systematic studies showing that heterozygous LOF APOC2 carriers have moderate HTG.

LMF1: The LMF1 gene, located on human chromosome 16p13.3, encodes the lipase maturation factor 1 that is integral to the proper folding and maturation of several lipases: lipoprotein lipase (LPL), hepatic lipase (HL), and endothelial lipase (EL).

Because of its unique role in activating multiple lipases, bi-allelic genetic LOF LMF1 variants cause combined lipase deficiency (CLD). Clinically, CLD is difficult to differentiate from FCS. People with CLD may also present with disturbances in phospholipid metabolism due to failure to activate endothelial lipase, but this is rarely screened for in the clinic.

To date, there is no systematic evidence from large biobanks showing that heterozygous LMF1 pathogenic variant carriers have moderate hypertriglyceridemia.

GPIHBP1: The GPIHBP1 gene, located on human chromosome 8q24.3, encodes the glycosylphosphatidylinositol-anchored high-density lipoprotein-binding protein 1 that serves as a molecular anchor for lipoprotein lipase, positioning it within the endothelial luminal wall.

To date, there is no systematic evidence from large biobanks showing that heterozygous GPIHBP1 pathogenic variant carriers have moderate hypertriglyceridemia.

### 3.1. Rare Diseases Presenting with Hypertriglyceridemia

Familial dysbetalipoproteinemia (FDBL): Also known as type 3 hyperlipoproteinemia, FDBL primarily occurs due to homozygosity of the APOE2 allele. However, only 15–20% of E2 homozygotes develop the FDBL phenotype; other genetic, hormonal, dietary, and metabolic dysfunction interacts with the E2/E2 genotype to cause FDBL. Rare, loss-of-function APOE variants can cause autosomal dominant FDBL.

FDBL is clinically characterized by moderate-to-severe hypertriglyceridemia, hypercholesterolemia that may phenocopy familial combined hyperlipidemia (FCHL), hepatosplenomegaly, xanthomata, and a very high risk of coronary and peripheral artery disease. In some cases, FDBL can occur in people who are heterozygous for APOE2, but there are likely other genetic and perhaps non-genetic factors contributing to the pathogenesis and phenotype.

Importantly, FDBL is the exception regarding the close alignment of plasma apoB and LDL-C levels. The work by Professor Allan Sniderman and colleagues show that, in people with FDBL, plasma apoB levels are normal to low compared to cholesterol levels. This is because FDBL is fundamentally a disease of cholesterol-rich lipoprotein remnants. In FDBL, the absolute number of lipoprotein particles is not appreciably elevated, but rather the lipid cargo cannot be cleared by hepatic receptors. As such, people with FDBL will have a normal-to-low total cholesterol/apoB ratio [75,76].

Partial lipodystrophy: Lipodystrophies are a heterogeneous group of diseases characterized by aberrant distribution of fat tissue, insulin resistance, polycystic ovary syndrome, acanthosis nigricans, hepatosteatosis, hypertriglyceridemia, and, in some cases, hypercholesterolemia and associated conditions. The severity of abnormal fat distribution differs amongst the subtypes of lipodystrophy. Broadly, there are acquired vs. genetic/familial forms and generalized vs. partial forms of lipodystrophies.

Due to the limited scope of this review being focused on the genetic causes of dyslipidemias, this section will focus on familial partial lipodystrophies (FPLDs) to provide some basis for the differential diagnosis of sHTG. People with FPLD often present with sHTG and acute pancreatitis, and generally, the aberrant fat distribution is mild [77], which may be missed on a clinical exam. Internal, unpublished data of people with sHTG referred for GBinsight comprehensive genetic testing show that the amount diagnosed with FPLD outnumbers those diagnosed with FCS by 3:1. Importantly, as treatments for FCS are receiving regulatory approval and clinical trials for FPLD are underway, it is imperative that these rare diseases are appropriately diagnosed. We refer the reader interested in a deeper perspective on lipodystrophies for clinicians by Patni and Garg [78].

FPLD2: Also known as Dunnigan-type partial lipodystrophy after the Scottish physician Matthew Dunnigan, FPLD2 is due to rare missense variants in the LMNA gene and is inherited in an autosomal dominant manner. The LMNA gene, located on human chromosome 1q22, encodes for two proteins, lamin A and lamin C, which arise due to the alternative splicing of the last exon. These proteins make up the nuclear lamina that envelopes the cell’s nucleus and provides structural support as well as regulates gene expression programs via the spatial arrangement of the genome. This epigenetic regulation is thought to allow for adipocyte differentiation and identity.

Most FPLD2-causing pathogenic genetic variants occur within exon 8 or 11 of the LMNA gene, most notably affecting arginine482 and arginine582 [79]. These pathogenic variants are known to affect adipocyte function.

FPLD3: The most common form of FPLD is due to loss-of function variants in the PPARG gene that is inherited in an autosomal dominant manner. The PPARG gene, located on human chromosome 3p25.2, encodes the peroxisome proliferator-activated receptor gamma transcription factor that is a master regulator of white adipose tissue differentiation and function.

People with FPLD3 may present with milder lipodystrophy but have severe metabolic dysfunction. This may be one reason why people with FPLD3 may be underdiagnosed, as they can more readily be suspected of having FCS or related endocrinopathy [77].

Other FPLD subtypes: FPLD4 is caused by loss-of-function variants in the PLIN1 gene that is inherited in an autosomal dominant manner. PLIN1 encodes for perilipin-1 that is highly expressed in white adipocytes that coats lipid storage droplets and serves to regulate lipolysis and therefore triglyceride homeostasis.

FPLD6 is caused by loss-of-function variants in the LIPE gene that is inherited in an autosomal recessive manner, which makes this subtype unique. The LIPE gene encodes for hormone-sensitive lipase that is involved in the lipolysis of stored triglycerides in white adipocytes for energy utilization. This enzyme is acutely responsive to the actions of insulin; catecholamines induce activation of this enzyme. Heterozygous loss-of-function carriers of pathogenic LIPE variants have moderate hypertriglyceridemia and an increased risk of type 2 diabetes [80].

There are other much rarer FPLD subtypes that are still being reported, but few of these have been replicated, to date.

Glycogen Storage Diseases (GSDs): Glycogen storage diseases are a group of monogenic disorders that involve defects in glycogen synthetic and catabolic processes in the liver and/or muscle. As a general rule, GSDs are inherited in an autosomal recessive manner. A more thorough review of these heterogeneous disorders can be found elsewhere [81]. This abbreviated section will focus on GSD subtypes that cause primary hypertriglyceridemia. Glycogen storage disease type IA (GSD1A) is the prototypical example.

Glycogen storage disease type 1 (GSD1): Also known as von Gierke disease, GSD1A is due to bi-allelic loss-of-function variants in the G6PC1 (also known as G6PC) gene. GSD1A is the most common subtype of GSD. Clinically, GSD1A may result in hypertriglyceridemia, xanthomas, hypoglycemia, hyperuricemia, lactic acidosis, and hepatosteatosis. Hypertriglyceridemia may be in the severe range. The age of onset is within the first year of life.

The G6PC1 gene encodes the catalytic subunit of the glucose-6-phosphatase (G6Pase) enzyme that converts glucose-6-phosphate (G6P) to glucose as part of the gluconeogenesis and glycogenolysis pathways. In the absence of G6Pase activity, there is an accumulation of glycogen and G6P. Excess G6P in the liver, through glycolysis, gets converted to fatty acids via the DNL pathway, which further promotes VLDL production [82].

Heterozygous carriers of pathogenic G6PC1 variants have moderate hypertriglyceridemia, according to gene burden analysis from the UKBiobank [62].

Pathogenic loss-of-function variants in other genes that enable G6Pase enzyme activity also cause GSD type 1 in an autosomal recessive manner. For example, the SLC37A4 gene encodes a protein with G6P translocase activity that transports G6P from the cytoplasm to the endoplasmic reticulum. Genetic loss of this activity leads to the accumulation of G6P, and the clinical sequelae is known as GSD1B. Heterozygous carriers of loss-of-function SLC37A4 variants may have mild hypertriglyceridemia.

Treatment for GSD1 is focused on preventing hypoglycemia with regular feedings of corn starch. Lipid-lowering therapies may be necessary to control hypertriglyceridemia [82].

Other rarer GSD subtypes causing hypertriglyceridemia: Also known as Cori or Forbes disease, GSD3 is an autosomal recessive disease caused by loss-of-function variants in the AGL gene. This gene encodes for the glycogen debranching enzyme that is involved in glycogenolysis. Clinical presentation of GSD3 is similar to GSD1 in some regards with hypertriglyceridemia, hypoglycemia, hyperuricemia, hepatomegaly, etc. Data from the UKBiobank do not show moderate hypertriglyceridemia in heterozygous carriers of loss-of-function AGL genetic variants.

There are approximately 20 GSD subtypes caused by different genes involved in glycogen metabolism and blood sugar homeostasis. Each of these has a different clinical presentation, but only a few of them, notably GSD1, are associated with hypertriglyceridemia. Importantly, treatment is distinct from other primary causes of hypertriglyceridemia.

### 3.2. Non-LPL Pathway Causes of Hypertriglyceridemia

GCKR: The GCKR gene, located on human chromosome 2p23.3, encodes the glucokinase regulatory protein (commonly referred to as GKRP) that serves to sequester the glucokinase (GCK) enzyme from the cytosol and prevent phosphorylation of hexoses. This sequestration occurs in pancreatic islet cells and hepatocytes with the goal of regulating the rate-limiting step in glucose utilization and effectively functions as an intracellular glucometer. When GCK is active, glucose uptake occurs, while during GKRP sequestration of GCK, glucose uptake is limited. Fructose and its metabolites can modulate the non-covalent interaction between GCK and GKRP [83].

Genome-wide association studies have consistently identified a missense variant, p.Leu446Pro (L446P; rs1260326), associated with elevated triglyceride levels, higher liver fat content, increased risk of metabolic dysfunction-associated fatty liver disease (MAFLD), and higher uric acid levels [84,85,86,87,88]. At the same time, this variant is associated with reduced fasting blood glucose levels and reduced type 2 diabetes risk [89,90]. This variant and other rare, loss-of-function GCKR variants are unable to sequester glucokinase, which enhances glucose uptake and lowers plasma glucose levels but increases intrahepatic glucose concentrations. This promotes de novo lipogenesis (DNL), which increases very low-density lipoprotein (VLDL) production and secretion, which increases the risk of hypertriglyceridemia [91]. Rare, genetic loss-of-function GCKR variants have been identified in people with clinically relevant hypertriglyceridemia [92,93].

CREB3L3: The CREB3L3 gene, located on human chromosome 19p13.3, encodes the cAMP-responsive element-binding protein, hepatic-specific (CREB-H) transcription factor that is involved in regulating triglyceride and glucose metabolism. CREB-H directly controls the expression of several genes involved in triglyceride metabolism, such as APOC2, APOA4, and APOA5 [94].

Data from the UKBiobank cohort shows that genetic loss-of-function CREB3L3 burden analysis is associated with robust increases in serum triglyceride levels [62,95]. One genetic variant, c.732dup(p.Lys245GlufsTer130), that results in a frameshift and premature truncation of the CREB-H protein has been consistently found in people with hypertriglyceridemia [94,95,96].

### 3.3. Polygenic Hypertriglyceridemia

Most cases of moderate-to-severe hypertriglyceridemia are due to polygenic causes [97]. Multifactorial chylomicronemia syndrome (MCS) is the major cause of sHTG that is associated with a high risk of acute pancreatitis, ASCVD, type 2 diabetes, and MAFLD [98]. MCS is characterized by defects in the clearance of triglyceride-rich lipoproteins due to the limited lipoprotein lipase activity as well as enhanced VLDL production/secretion. MCS is due to polygenic causes but can also occur due to heterozygous carriers of rare, pathogenic variants in the LPL and APOA5 genes. But it likely requires other genetic and non-genetic factors to cause sHTG. MCS is relatively frequent with an estimated prevalence in the United States of 1:250–1:600 [98].

With the approval of the anti-APOC3 agent, olezarsen, for FCS and the prospect of anti-ANGPTL3 agents in late-stage clinical trials, it is becoming increasingly evident that sHTG has divergent causes, and knowing the proximal cause will inform the best treatment options. The comprehensive genetic screening of the multitude of causes offers to provide a roadmap for precise therapies for sHTG.

## 4. High-Density Lipoprotein (HDL)/Reverse Cholesterol Transport Pathway

Background: Epidemiological studies consistently show a U-shaped curve between HDL-cholesterol levels and all-cause and cardiovascular disease mortality [99,100]. Genetic epidemiology studies employing mendelian randomization to estimate the lifelong exposure to low or high HDL-cholesterol levels are mixed with regards to ASCVD risk [101]. One explanation is that these studies differ in genes included in their models.

Studies with pharmacological interventions to increase HDL-cholesterol levels have, to date, not shown any cardioprotective effects [102]. Recently, the infusion of recombinant apoA1 (CSL112) in people with existing ASCVD did not reduce risk of major adverse cardiovascular events (MACEs) within 3 months [103], further raising doubt about the cardioprotective effects of HDL.

Perhaps, assessing reverse cholesterol transport function instead of static HDL-C biomarkers is informative. A study evaluating cholesterol efflux capacity, a measure of HDL function, showed an inverse relationship: higher efflux capacity was associated with 67% reduced ASCVD risk [104]. Conventional metrics of HDL-C were not associated with risk. These findings were replicated in subsequent independent cohorts [105,106].

There is still much to be learned about HDL, reverse cholesterol transport, and the risk of complex cardiometabolic diseases. Beyond ASCVD risk, there is some preliminary evidence showing protective effects of HDL and higher HDL-C and reduced risk of type 2 diabetes [107]. In this section, we discuss the genetics of low and high HDL-C and its effects on ASCVD risk.

### 4.1. Hypoalphalipoproteinemia

APOA1: Genetic loss-of-function variants in the APOA1 gene cause familial hypoalphalipoproteinemia in an autosomal co-dominant manner. Heterozygous hypoalphalipoproteinemia causes moderately lower HDL-cholesterol levels and increases ASCVD risk. Data from the UKBiobank shows that heterozygous carriers of pathogenic APOA1 variants also have moderately higher serum triglyceride levels [62]. Bi-allelic hypoalphalipoproteinemia causes near complete absence of serum HDL-C and apoA1 that causes corneal opacities, xanthoma, and ASCVD. As such, there is a dose–response relationship between apoA1, HDL-C, and the ASCVD risk. Further supporting a role for APOA1 in ASCVD risk, a rare genetic variant, p.Val43Leu, that is found in nearly 1% of people in Iceland is associated with a marked increase in HDL-C and is associated with reduced ASCVD risk [108]. This variant is a likely gain-of-function allele.

Some rare, pathogenic APOA1 variants can also cause hereditary systemic amyloidosis in an autosomal dominant manner. These variants alter the shape of the apoA1 protein that leads to misfolding and systemic amyloid deposition. This causes toxicities and can affect the tissue/organ where amyloid deposits occur. APOA1 amyloidosis preferentially affects the heart, skin, and peripheral neurons.

The APOA1 gene, located on human chromosome 11q23.3, encodes for the apoA1 protein that is produced by hepatocytes and enterocytes and secreted into circulation as pre-beta HDL (also known as discoidal HDL). The nascent HDL matures as it picks up peripheral cholesterol and phospholipids as part of the reverse cholesterol transport pathway. In addition to serving a structural role on HDL, apoA1 activates the LCAT enzyme to form stable cholesterol esters. Once mature, apoA1 will bind to hepatic scavenger receptors, particularly SR-B1, to facilitate clearance of this cholesterol-rich HDL. ApoA1 HDL is then recycled back to the nascent HDL stage.

ABCA1: Heterozygous genetic loss-of-function variants in the ABCA1 gene causes familial hypoalphalipoproteinemia; bi-allelic loss-of-function ABCA1 causes Tangier disease. Heterozygous ABCA1 loss-of-function variant carriers have moderate-to-severe hypoalphalipoproteinemia with increased ASCVD risk. Tangier disease leads to severe hypoalphalipoproteinemia, corneal opacities, enlarged, yellow tonsils, hepatomegaly, splenomegaly, and premature ASCVD.

The ABCA1 gene, located on human chromosome 9q31.1, encodes the ATP-Binding Cassette A1 protein that is involved in pumping cholesterol out from peripheral cells onto nascent HDL. ABCA1 also exchanges phospholipids between lipoproteins and cells.

ABCA1 is also involved in the clearance of amyloid beta and in the lipidation of apoE within the brain [109]. Genetic studies show that rare ABCA1 genetic variants are associated with an increased risk of Alzheimer’s disease dementia [110].

LCAT: LCAT deficiency is an autosomal recessive condition characterized by severe hypoalphalipoproteinemia, hypertriglyceridemia, and corneal opacities. LCAT deficiency can be categorized based on residual enzyme activity, which tracks disease severity. Fish-eye disease is the less severe subtype due to some residual LCAT activity. Norum disease is the more severe disease that typically includes renal involvement. Heterozygous LCAT genetic loss-of-function variant carriers will have moderately lower HDL-C and moderate hypertriglyceridemia.

The LCAT gene, located on human chromosome 16q22.1, encodes the lecithin-cholesterol acyltransferase enzyme that is involved in the esterification of cholesterol esters within the core of HDL particles. This reaction enables the packing and therefore maturation of HDL particles.

At least one genetic epidemiology study of Northern Europeans did not find an increased risk of ASCVD in those carrying an LCAT genetic variant associated with reduced HDL-C levels [111].

### 4.2. Other Genes Associated with Low HDL-C

PLTP: The PLTP gene, located on human chromosome 20q13.12, encodes the phospholipid transfer protein that is involved in the exchange of cholesterol and phospholipids from VLDL and LDL to HDL particles, and via this reaction, PLTP helps HDL particles mature.

GWAS show that common genetic variants (e.g., rs6065906) near and within the PLTP gene are associated with reduced HDL-C and higher triglycerides [51,63,85]. Gene burden analysis from the UKBiobank shows that rare, loss-of-function PLTP genetic variants are associated with reduced HDL-C [62]. Further, a rare missense variant, p.Val70Phe, in the PLTP gene that is enriched in people of Finnish European ancestry is associated with a three-fold increase in coronary heart disease risk [112]. Model rodent experiments consistently show that the inactivation of the PLTP gene reduces HDL particle size and HDL-C levels [113]. However, to date, there is no recognized monogenic disease assigned to PLTP genetic deficiency.

### 4.3. Hyperalphalipoproteinemia

CETP: Genetic loss-of-function CETP variants cause hyperalphalipoproteinemia 1, which is inherited in an autosomal co-dominant manner. People with CETP loss-of-function variants have elevated HDL-C, lower apoB, and LDL-C. People of East Asian ancestry have the highest rates of hyperalphalipoproteinemia due to CETP loss-of-function with the p.Asp459Gly variant (D459G, although older nomenclature assigned this variant, D442G) being found in 5% of people of East Asian ancestry.

While earlier studies did not show cardioprotection in people of East Asian ancestry given the high frequency of the D459G variant [114,115], larger cohorts that include non-East Asians have shown reduced ASCVD risk in people with CETP loss-of-function genetic variants [116,117]. A likely explanation for this difference is that the D459G variant does not appreciably affect apoB or LDL-C levels [115]. This consideration is a focus of a possible renaissance of CETP inhibitors for reducing ASCVD risk [118].

The CETP gene, located on human chromosome 16q13, encodes the cholesteryl ester transfer protein that is involved in the equimolar exchange of cholesteryl esters from HDL to VLDL and LDL particles, while triglycerides flow from VLDL to HDL particles.

*APOC3:* Genetic loss-of-function APOC3 variants are known to cause hyperalphalipoproteinemia 2 in an autosomal co-dominant manner. Hyperalphalipoproteinemia 2, also known as APOC3 deficiency, is characterized by moderate-to-very high HDL-C and low triglyceride levels. APOC3 deficiency is consistently associated with reduced ASCVD risk [119,120].

The APOC3 gene, located on human chromosome 11q23.3, encodes the apoC-III protein that serves as an endogenous inhibitor of lipoprotein lipase activity. ApoC-III is produced by hepatocytes and associates with most lipoprotein subtypes. Besides inhibiting lipoprotein lipase, apoC-III plays a role in the secretion of VLDL as well as slowing its catabolism. As such, apoC-III plays an important role in metabolism of triglyceride-rich lipoproteins. Genetic APOC3 loss-of-function therefore enhances lipoprotein lipase-mediated clearance of TRLs, and by virtue of lower triglyceride levels, increases HDL-C.

Hepatic lipase deficiency: Genetic loss-of-function variants in the LIPC gene causes hepatic lipase deficiency in an autosomal recessive manner. Hepatic lipase deficiency is characterized by hyperalphalipoproteinemia, hypertriglyceridemia, moderate hypercholesterolemia, xanthomas, increased risk of ASCVD, and may cause hepatomegaly. Heterozygous carriers of loss-of-function LIPC variants may have moderately high HDL-C and moderate hypertriglyceridemia.

GWAS consistently show that common SNPs within the LIPC gene, notably within the promoter region (-250G>A, rs2070895), are associated with HDL-C [121,122].

The LIPC gene, located on human chromosome 15q21.3, encodes hepatic lipase (HL) that is involved in the catabolism of triglyceride-rich lipoproteins by hydrolyzing triglycerides in these lipoproteins and helping to convert VLDL to IDL and to LDL. Separately, hepatic lipase aids in the clearance of HDL particles by facilitating the binding of apoA1-HDL particles to the SR-B1 hepatic receptor.

SCARB1: The SCARB1 gene, located on human chromosome 12q24.31, encodes the hepatic scavenger receptor B1 (SR-B1) that is the major means of receptor-mediated HDL clearance by the liver. Consistently, people with loss-of-function SCARB1 genetic variants are known to have increased HDL-C levels [95,123]. In some studies, people with SCARB1 loss-of-function variants have increased ASCVD risk [124].

There is some preliminary evidence that SR-B1 serves as a receptor for clearing Lp(a) from circulation [125,126]. In some studies, people with SCARB1 genetic variants have high Lp(a) levels [127]. If this proves correct, this would implicate SCARB1 genetic variants as another strong ASCVD risk factor, beyond HDL dysfunction.

LIPG: The LIPG gene, located on human chromosome 18q21.1, encodes the endothelial lipase (EL) enzyme. This enzyme is involved in the hydrolysis of phospholipids and has particular affinity for HDL [128,129]. Functionally, EL is involved in the degradation of HDL particles via its phospholipase activity.

GWAS and large biobank studies show that common and rare genetic LIPG variants are associated with HDL-C with rare loss-of-function variants associated with moderately increased HDL-C [130]. However, genetic studies do not support an association between LIPG loss-of-function variants and the ASCVD risk [131,132].

HDL-C levels ascertained from a lipid panel are likely insufficient information to help with ASCVD risk assessment. HDL functional data, such as cholesterol efflux capacity results, would be superior, but this test is not readily ordered in the clinic. Comprehensive genetic analysis can provide some granularity about ASCVD risk by identifying the genetic variant(s) and genes responsible for an individual’s low or high HDL-C. As noted above, not all genetic causes of hyperalphalipoproteinemia are associated with increased ASCVD risk. Knowing the proximate genetic cause is informative for risk assessment.

## 5. Conclusions

Hyperlipidemia, particularly hypercholesterolemia, is a causal risk factor for ASCVD. Hypertriglyceridemia is likely a risk factor. These risk factors are highly heritable with estimates between 40 and 60% [133,134]. There is a multitude of genetic causes contributing to lipid levels ranging from very rare (e.g., sitosterolemia) to common (e.g., polygenic hypercholesterolemia). But in the primary care setting with only a serum lipid panel, the precise cause can be uncertain. Knowing the proximate cause has clinical utility.

There is additive ASCVD risk when an individual has multiple risk factors, such as familial hypercholesterolemia with high Lp(a) [135]. Comprehensive genetic analysis can delineate whether a person with hypercholesterolemia and high Lp(a) is due to monogenic FH and high Lp(a), which carries a very high ASCVD risk, versus someone with high Lp(a) that presents as clinical hypercholesterolemia.

Precision medicine with comprehensive genetic analysis offers improved clinical outcomes by selecting the most appropriate treatments for a particular disease, while also minimizing unnecessary adverse effects. Figure 2 presents an infographic that summarizes some of the conditions that phenocopy other conditions that may share lipid abnormalities but have different treatments.

As shown in Figure 2, treatment is different for people with sitosterolemia due to ABCG5/ABCG8 pathogenic variants that cause severe hypercholesterolemia and ASCVD and can phenocopy FH. First-line therapy for people with sitosterolemia is ezetimibe. However, in people with FH, ezetimibe is ancillary to diet and maximally tolerated statins.

The utility of genetics-guided therapies is highlighted by the recent approval of olezarsen for the treatment of FCS. The APOC3 antisense oligonucleotide, olezarsen, was approved for the treatment of FCS on top of diet therapy [136]. Olezarsen is not indicated for people with familial partial lipodystrophy, which can phenocopy FCS. Clinical trials are exploring other therapeutic options for people with FPLD.

Comprehensive genetic analysis that is agnostic of the condition, but takes a phenotype-centric approach, or whole exome/whole genome sequencing, promises to guide true precision medical interventions.

Limitations: The price of genetic testing is rapidly dropping, making comprehensive genetic analysis more accessible. However, insurance coverage is still a major obstacle due to the perception of its limited clinical utility. As noted, there is clear clinical utility for the ascertainment of whether an individual’s clinical FH is monogenic or polygenic [6,7,8,9]. Older studies have shown that genetic testing for FH is cost-effective [137,138]. And these analyses were performed when the cost of next-generation sequencing was more expensive than it is today.

There are other reasons why people do not wish to know their genetics that involve fears of violations of privacy and being dropped by health and/or life insurance policies. The Genetic Information Nondiscrimination Act (GINA), signed into law in 2008, prohibits health insurance companies and employers from discriminating against someone based on genetic findings. However, life insurance companies, to date, are not covered by this law and can deny coverage based on genetic findings. Some states have additional laws to help protect against genetic discrimination in healthcare.

## Figures and Tables

**Figure 1 nutrients-17-00659-f001:**
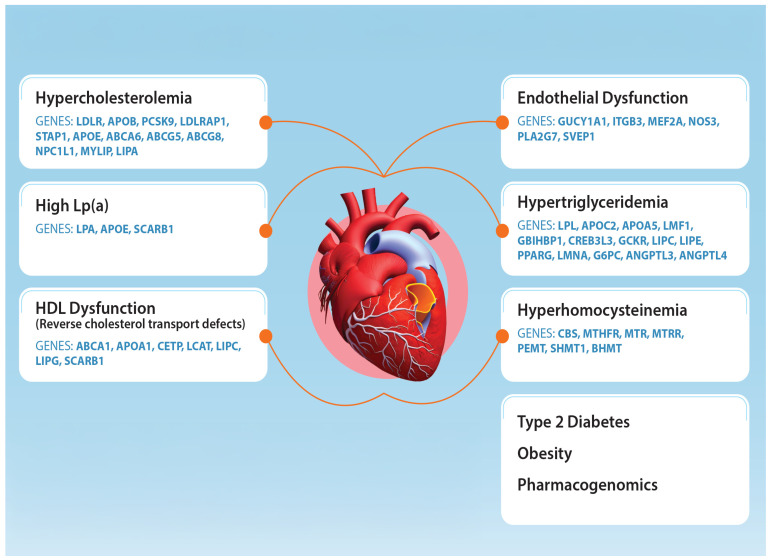
This diagram shows the multitude of risk factors contributing to the ASCVD risk and a selection of genes contributing to these risk factors. The GBinsight Comprehensive Dyslipidemia and ASCVD genetics panel screens these genes to provide a comprehensive analysis of ASCVD risk in a single genetic test.

**Figure 2 nutrients-17-00659-f002:**
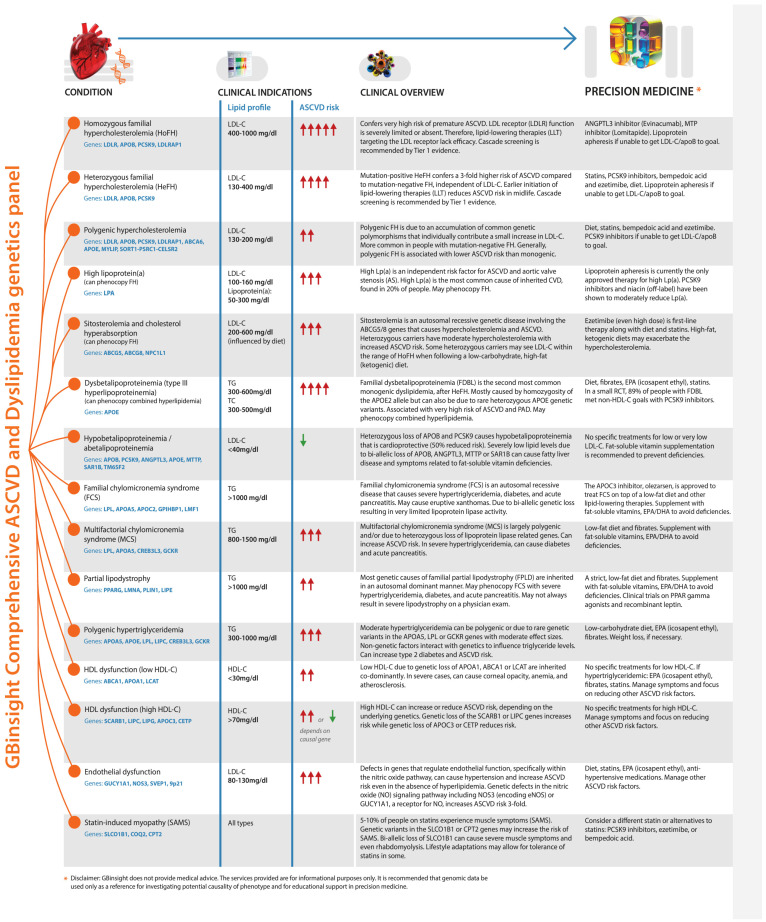
Infographic summarizing different conditions that cause primary dyslipidemias and their relative effect on ASCVD risk. When applicable, guidelines-based recommendations are suggested for the specific condition. Red upward-facing arrows represent increased ASCVD risk. Green downward-facing arrows represent relative cardioprotection. The number of arrows is proportional to relative risk of ASCVD for a given condition. When red and green arrows co-occur in the same condition, it indicates that ASCVD risk can be either higher or lower depending on the genetic cause(s). A high-resolution version of this figure can be downloaded at the following URL: https://www.gbhealthwatch.com/downloads/GBinsight%20Precision%20Medicine-poster1.pdf (accessed on 5 February 2025).

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
