# Peer review of "A Comprehensive Review of the Genetics of Dyslipidemias and Risk of Atherosclerotic Cardiovascular Disease"

_nutrients, 2025, doi:10.3390/nu17040659_

Round 1

Reviewer 1 Report

Comments and Suggestions for Authors

### Review of the Article: "A Comprehensive Review of the Genetics of Dyslipidemias and Risk of Atherosclerotic Cardiovascular Disease"

---

#### General Comments
This article addresses a highly relevant and timely topic: the genetic underpinnings of dyslipidemias and their contribution to atherosclerotic cardiovascular disease (ASCVD) risk. Given the increasing focus on precision medicine, the authors’ emphasis on the utility of comprehensive genetic analysis is laudable. The article provides a wealth of information, covering both well-established and emerging genetic contributors to dyslipidemia. However, the manuscript has notable areas that require attention to improve clarity, organization, and scientific rigor.

---

#### Strengths
1. **Comprehensive Scope**:
   The article thoroughly reviews canonical and non-canonical genetic causes of dyslipidemia, encompassing monogenic and polygenic contributors. It also discusses rare genetic conditions, highlighting their diagnostic and therapeutic implications.

2. **Focus on Precision Medicine**:
   The authors effectively underscore the importance of genetic analysis in tailoring treatments for dyslipidemias, aligning with current trends in personalized healthcare.

3. **Visual Representation**:
   The use of figures, such as pathway diagrams and infographics, aids in summarizing complex information.

4. **Clinical Relevance**:
   The discussion of treatment strategies tailored to specific genetic conditions adds a practical dimension to the review, making it valuable for clinicians.

---

#### Weaknesses and Areas for Improvement

1. **Clarity and Readability**:
   - The manuscript is dense and overly technical in parts, which may overwhelm readers unfamiliar with advanced genetics. The writing would benefit from clearer organization and simplification of complex concepts, particularly in sections with long paragraphs and technical terminology.
   - Sentences are occasionally verbose and redundant (e.g., in the introduction and conclusion). Streamlining the text would enhance readability.

2. **Structure and Organization**:
   - The review would benefit from a more structured format. For example:
     - The introduction could provide a clearer outline of the manuscript's scope.
     - The discussion of hypercholesterolemia, hypertriglyceridemia, and HDL abnormalities is detailed but lacks a systematic approach. Using subsections with standardized headings (e.g., "Genetic Basis," "Clinical Implications," and "Therapeutic Strategies") would improve flow and make the article easier to navigate.
     - The conclusion could be expanded to include key take-home messages and future directions for research and clinical practice.

3. **Lack of Critical Analysis**:
   - The article primarily summarizes existing knowledge without critically appraising the evidence. For example, while the authors mention discrepancies in the association between HDL-C levels and ASCVD risk, a deeper exploration of conflicting studies and their implications would strengthen the discussion.
   - The manuscript would benefit from a clearer distinction between well-established findings and areas of ongoing investigation.

4. **Figure Quality and Description**:
   - The figures, though helpful, lack detailed captions and clear references in the text. For instance, Figure 1 and Figure 2 are not adequately explained, limiting their utility.

5. **Citation and Referencing**:
   - Some claims lack proper citation, particularly in sections discussing novel genetic variants and their clinical significance. Adding references to primary literature would enhance credibility.

6. **Abstract and Keywords**:
   - The abstract is vague and could better summarize the article's key findings and conclusions. It does not provide sufficient detail to engage readers or convey the manuscript’s scientific contribution.
   - The keywords are exhaustive but could be condensed for relevance.

7. **Technical Errors**:
   - There are grammatical issues, such as awkward phrasing ("...may or not include Lp(a) or apoB"), inconsistent use of tenses, and typographical errors ("As will see" instead of "As we will see").
   - The authors use inconsistent gene nomenclature and variant descriptions (e.g., LDLR vs. LDL receptor, c.137T>C vs. p.Leu46Pro). Adhering to standardized nomenclature would improve clarity.

8. **Depth in Rare Disorders**:
   - While the article admirably covers rare genetic causes of dyslipidemia, some conditions (e.g., familial partial lipodystrophy) are discussed in excessive detail, potentially distracting from the main focus on ASCVD risk.

9. **Missing Discussion on Limitations**:
   - The article does not address limitations of genetic testing, such as cost, accessibility, and interpretation challenges. Including this perspective would provide a more balanced view.

---

#### Specific Comments and Recommendations

1. **Abstract**:
   - Revise to succinctly highlight the article's aims, key findings, and implications for clinical practice. Avoid vague statements.

2. **Introduction**:
   - Clarify the article's objectives and provide a roadmap for readers.

3. **Main Text**:
   - Use consistent subheadings and structure (e.g., "Background," "Mechanisms," "Clinical Implications," and "Future Directions").
   - Provide more critical evaluation of the evidence, including gaps in knowledge and areas for future research.

4. **Figures**:
   - Enhance figure captions to make them self-explanatory.
   - Ensure all figures are clearly referenced in the text.

5. **Conclusion**:
   - Expand to summarize key findings, emphasize the clinical importance of genetic testing, and outline research priorities.

6. **Editing and Proofreading**:
   - Address grammatical and typographical errors.
   - Standardize nomenclature and terminology.

---

#### Conclusion

This manuscript provides an extensive and valuable review of the genetics of dyslipidemias and ASCVD risk. However, to meet the standards of a high-impact scientific journal, it requires substantial revisions in organization, clarity, critical analysis, and technical accuracy. The authors are encouraged to simplify complex sections, adopt a systematic structure, and critically appraise the literature to provide a balanced and insightful review. With these improvements, the article has the potential to make a meaningful contribution to the field.

Author Response

Comment 1: Revise Abstract to succinctly highlight the article's aims, key findings, and implications for clinical practice. Avoid vague statements.

Response 1: The abstract was rewritten to provide a more clear and succinct summary of the major points of this review. 

Comment 2:   Clarify the article's objectives and provide a roadmap for readers.

Response 2: At the end of the Introduction, we provide a roadmap for readers. 

Comment 3: Use consistent subheadings and structure (e.g., "Background," "Mechanisms," "Clinical Implications," and "Future Directions").
   - Provide more critical evaluation of the evidence, including gaps in knowledge and areas for future research.

Response 3: We added subheadings within the three broad sections of the main body. 

With regards to a specific section on clinical applications, we believe this is better placed within the relevant subsections. For example, when discussing sitosterolemia (line 125), we note within this subsection that ezetimibe is firstline therapy (line 132). We organized this article that is centered on genetics to be arranged by genetic condition. In developing the concept for this review, we imagine what an internal medicine physician, armed only with a patient's lipid panel, would face in trying to differentially diagnose a type of dyslipidemia and how does this information influence this patient's risk of ASCVD. And knowing this information, what types of treatments are available. With this guide in mind, we have arranged the subsections. 

As for critical evaluation of the studies, we do feel, given the wide scope of this review, we spent the appropriate amount of space discussing this. In particular, in the section on HDL, we review in general terms, the conflicting evidence linking HDL dynamics to ASCVD risk (line 505). We added a critical analysis of the study concluding that aspirin may reduce ASCVD risk in people with high Lp(a) (line 189). 

Comment 4: The figures, though helpful, lack detailed captions and clear references in the text. For instance, Figure 1 and Figure 2 are not adequately explained, limiting their utility.

Comment 4: Figure 1 is referenced in the Introduction section of the review as a way to introduce the concept that ASCVD is multifactorial and a comprehensive genetics panel that is focused on the myriad of pathways contributing to ASCVD risk and therefore allowing for additive risk analysis and precise diagnostics can unlock the mysteries of a patient's relative ASCVD risk. Figure 2 is referenced in the Conclusion and serves as a quick guide for physicians without a deep understanding of the different genetic conditions to use genetic findings to accurately diagnose their patients and guide therapies accordingly. We added a URL within the figure 2 header so the reader can download a full version of this figure (line 681). 

Comment 5:  - Some claims lack proper citation, particularly in sections discussing novel genetic variants and their clinical significance. Adding references to primary literature would enhance credibility.

Response 5: We are unsure of what the reviewer is referring to. When mentioning a specific genetic variant, we include references (for example see lines 111, 120, 189, 263, 391, 486 etc). The reviewer states "...in sections discussing novel genetic variants and their clinical significance..." we dont mention any novel variants or the clinical significance of any novel variants. This review discusses scientific consensus. 

Comment 6: The abstract is vague and could better summarize the article's key findings and conclusions. It does not provide sufficient detail to engage readers or convey the manuscript’s scientific contribution.
   - The keywords are exhaustive but could be condensed for relevance.

Response: We updated the abstract to be more informative. We updated the keywords to be more condensed. 

Comment 7:   - There are grammatical issues, such as awkward phrasing ("...may or not include Lp(a) or apoB"), inconsistent use of tenses, and typographical errors ("As will see" instead of "As we will see").
   - The authors use inconsistent gene nomenclature and variant descriptions (e.g., LDLR vs. LDL receptor, c.137T>C vs. p.Leu46Pro). Adhering to standardized nomenclature would improve clarity.

Response 7: We appreciate the reviewers pointing out that we did not sufficiently proofread our draft. We have made substantial efforts to reread and proof our manuscript to correct these typos. 

With respect to specific variant nomenclature, our aim was, when appropriate, use commonly recognized variant names to make the reader more familiar. For example, when discussing APOE alleles, we prefer to use the commonly used E2, E3, E4 instead of the HGVS (Human Genome Variation Society)-accepted c.526C>T(p.Arg176Cys) to denote E2. Likewise, when referencing the most common form of APOB-defecting familial hypercholesterolemia variant, Arg3527Gln (line 69), this is a commonly referenced variant and the more familiar nomenclature is R3527Q/R3500Q as is noted. The HGVS-accepted variant name, c.10580G>A(p.(Arg3527Gln), is less likely to be recognized by the average reader. However, in some instances, we do include the HGVS and common nomenclature because these are less likely to be recognized by the average reader. For example, in discussing the APOE in-frame deletion variant, Leucine167 (c.500_502del(p.Leu167del)) (line 110) or CREB3L3 c.732dup(p.Lys245GlufsTer130) (line 484).

Comment 8: While the article admirably covers rare genetic causes of dyslipidemia, some conditions (e.g., familial partial lipodystrophy) are discussed in excessive detail, potentially distracting from the main focus on ASCVD risk.

Response 8: We respectfully disagree about excessive discussion dedicated to ultra rare diseases, such as partial lipodystrophy (FPLD). We include some unpublished data and a brief discussion on why we believe it is prudent to include FPLD within the subsection on phenocopying FCS and severe hypertriglyceridemia (line 374). We provide justification for bringing this topic into this discussion on dyslipidemias. Familial chylomicronemia syndrome (FCS) has been widely studied. Partial lipodystrophy is much less published and underdiagnosed, which is why we included this subsection with some granularity.  

There are many great review articles that have focused on individual diseases. As noted above, the concept for this article to present the myriad of causes of dyslipidemias that a non-specialist clinical may encounter. Our genetic testing service has returned clinical diagnoses to multiple clinicians in the primary care setting with rare diseases because the physician and patient lacks the expertise to properly diagnose these diseases. FPLD is a known cause of severe hypertriglyceridemia that may phenocopy FCS. As such, we feel this warrants being discussed as part of the comprehensive overview of the genetics of dyslipidemia. 

Comment 9: Missing Discussion on Limitations**:
   - The article does not address limitations of genetic testing, such as cost, accessibility, and interpretation challenges. Including this perspective would provide a more balanced view.

Response 9: We added a subsection within the Conclusion to include the reviewers recommendations. 

Reviewer 2 Report

Comments and Suggestions for Authors

This comprehensive review summarized the genetic causal genes that contribute to atherosclerotic cardiovascular disease. I have a few minor concerns:

1.        Figure 1, “Endothelial Dysfunction” was duplicated and redundant.

2.        Please check the entire manuscript carefully and cite related references, such as Line 334, Line 433, Line 527, etc.

3.        Line 34, please list the full name of FH here instead of later in Line 35.

4.        The conclusion part is too simple.

5.        Figure 2, this panel includes 140 genes. How about the other genes apart from those listed in Figure 2?

Author Response

Comment 1: Figure 1, “Endothelial Dysfunction” was duplicated and redundant.

Response 1: This figure was updated and duplicated text has been removed. 

Comment 2:  Please check the entire manuscript carefully and cite related references, such as Line 334, Line 433, Line 527, etc.

Response: We have gone through the entire manuscript again with a fine-toothed comb and corrected typos etc. 

Comment 3: Line 34, please list the full name of FH here instead of later in Line 35.

Response 3: Familial hypercholesterolemia was changed to be spelled out in lin 34 and abbreviated as FH in later sections. 

Comment 4: The conclusion part is too simple.

Response 4: We rewrote the conclusion section that summarizes the major points of the article and emphasizes several prominent examples of precision medicine that likely requires genetics to delineate that cause of disease. 

Comment 5:  Figure 2, this panel includes 140 genes. How about the other genes apart from those listed in Figure 2?

Response: We amended Figure 2 to remove this reference to 140 genes. The genes listed in Figure 2 are merely the major genes associated with said condition and is not intended to be an exhaustive list. There is no space in this figure to assign the full spectrum of genes to each condition. 

Response 4: 

Reviewer 3 Report

Comments and Suggestions for Authors

This review article provides a comprehensive overview of these pathways and examines the evidence linking them to the risk of atherosclerotic cardiovascular disease (ASCVD). Hyperlipidemia, particularly hypercholesterolemia, is a well-established causal risk factor for ASCVD, while hypertriglyceridemia is also likely to contribute to risk. Precision medicine, facilitated by advanced genetic analyses, enhances therapeutic efficacy by identifying the most suitable treatment for a specific disease while minimizing unnecessary adverse effects. 

The reviewer finds this review article to be well-written and offers the following comments: 

Major comments:

1.       In Figure 1, the "Endothelial Dysfunction" column is repeated. One instance should be removed. 

2.       In Line 542, there is an extra space between "9q31.1" and "encodes," which should be corrected. 

3.       Figure 2 is well-designed; however, the text is too small to read. The font size should be increased for improved readability. 

Author Response

Comment 1: In Figure 1, the "Endothelial Dysfunction" column is repeated. One instance should be removed. 

Response 1: Figure 1 has been amended to remove this error. 

Comment 2:  In Line 542, there is an extra space between "9q31.1" and "encodes," which should be corrected. 

Response 2: This and other typos have been corrected. 

Comment 3:  Figure 2 is well-designed; however, the text is too small to read. The font size should be increased for improved readability. 

Response 3:  We unfortunately are unable to make this figure of larger font size while containing it within a single page.